# Modeling Spectral Energy Shifts in Spatio-Temporal Graph Anomaly Detection

**Yilin Liu** [1]   **Hongchao Zhang** [1]   **Ahmad Taha** [2]   **Taylor T. Johnson** [1]   **Meiyi Ma** [1]

## Abstract

Graph anomaly detection methods aim to distinguish anomalous nodes. While prior methods characterize anomalies through increased variation in the spectral energy distributions, they overlook those that result in decreased variation, i.e., camouflaged anomalies that appear normal. We show that this type of anomaly persists across multiple datasets and remains undetectable by existing spectral approaches. To address this limitation, we propose a node-level spectral energy formulation that is fully compatible with message passing and enables the detection of camouflaged anomalies. Building on this formulation, we introduce an energy-aware graph learning framework that models spectral shifts through energy-driven message passing in both static and time-series graphs. Besides, our unified architecture extends to temporal settings without introducing specialized sequence modules, enabling efficient learning under long sliding windows. Extensive experiments on large-scale benchmarks demonstrate the effectiveness and scalability of our approach. Our code is available at https://github.com/AICPS-Lab/Spectral-Energy-Shifts-in-GAD.

## 1. Introduction

Network systems underpins critical application domains including financial transaction networks (Ma et al., 2021), e-commerce platforms, social graphs (Hooi et al., 2016), and cyber-physical infrastructures (Deng & Hooi, 2021). In these applications, anomalies such as fraudulent transactions, fake reviews, and malicious accounts do not occur in isolation, but propagate through connectivity, amplify their impact, and cause substantial economic losses and system failures (Liu et al., 2021; Deng & Hooi, 2021). Detecting such anomalies in graph-structured data, known as Graph Anomaly Detection (GAD), is therefore essential yet remains an open problem (Liu et al., 2022).

In GAD, existing methods can be broadly categorized into two directions: spatial and spectral (Tang et al., 2023). Spatial methods aim to enhance graph representation learning, including approaches (Li et al., 2019; Dou et al., 2020). On the other hand, spectral methods address the problem by designing different graph filters, AMNet (Chai et al., 2022), to capture anomaly patterns in the frequency domain. Among these works, the spectral formulation proposed in (Tang et al., 2022) characterizes node attribute anomalies with a high variation trend known as the "right-shift" phenomenon. This formulation shows potential to learn anomaly behavior and has inspired subsequent studies (Lin et al., 2024; Duan et al., 2023).

The spectral formulation, however, suffers from two key limitations, namely, limited scope and scalability. The formulation focuses on "right-shift", where anomalies exhibiting high variance in energy distribution, and overlooks camouflaged anomalies (Gao et al., 2023). These anomalies are embedded in the graph by imitating benign node behavior, resulting in high similarity to normal nodes and little deviation from the majority distribution (Dou et al., 2020). Such anomalies exist in public benchmark datasets but cannot be effectively detected by existing energy-based formulations. In our observations on the YelpChi (Rayana & Akoglu, 2015) dataset (Fig. 1), anomalies do not consistently follow the right-shift pattern. Among the 32 features, anomalies exhibit lower spectral energy than normal nodes in 26 features. This result indicates that these anomalies do not significantly deviate from normal nodes, thereby being overlooked by spectral methods. Moreover, graph-level spectral energy computation is costly, limiting scalability to large graphs. To address these challenges, we propose the Energy Graph Neural Network (EGNN), a node-level spectral GAD method. EGNN broadens the scope of spectral formulation to include camouflaged anomalies by i) quantifying them using the Rayleigh quotient and ii) leveraging the observed "*left-shift*" phenomenon to describe the low-variance trend caused by the anomalies. To improve scalability, we propose a local energy surrogate to approximate graph energy.

[1]College of Connected Computing, Vanderbilt University, Nashville, USA [2] Department of Civil and Environmental Engineering, Electrical and Computer Engineering, Vanderbilt University, Nashville, USA. Correspondence to: Meiyi Ma <meiyi.ma@vanderbilt.edu>.

*Proceedings of the $43^{rd}$ International Conference on Machine Learning*, Seoul, South Korea. PMLR 306, 2026. Copyright 2026 by the author(s).

While static detection relies on a single snapshot of spectral footprint, temporal detection relies on the trajectory of energy distributions. The spectral formulation naturally focuses on the changes in energy distribution, making it a promising representation in the temporal setting. In time-series GAD, various approaches span unsupervised and semi-supervised paradigms, and range from linear models to deep learning methods (Tuli et al., 2022; Ruff et al., 2018; Xu et al., 2021; Wu et al., 2022). Most existing methods decouple spatial and temporal modeling by stacking temporal modules on static graph backbones (Pareja et al., 2020). This class of frameworks has a large number of learnable parameters and suffers from poor data efficiency under label-scarce and highly imbalanced settings. In addition, their reliance on sliding windows makes performance sensitive to window length (Huang et al., 2023; Ma et al., 2024), and optimization instability (Pascanu et al., 2013) (e.g., vanishing or exploding gradients). For temporal graphs, EGNN avoids introducing additional temporal modules; instead, it models temporal dependencies via energy transformations and adaptive gating within sliding windows, yielding a data-efficient design that is less sensitive to window size. Our contributions are summarized as follows.

- We reveal a previously overlooked camouflaged anomaly pattern that appears across multiple public datasets, and characterize it with spectral formulation.

- We propose the EGNN, a node-level spectral-energy-based GAD method. It broadens the scope of spectral formulation to include camouflaged anomalies and enables energy-driven message passing through GNNs. We further mitigate the scalability issue by using local energy surrogates to approximate the graph energy.

- We generalize EGNN to time-series graphs by leveraging a local spectral surrogate to characterize changes over time and adaptive gating within slide windows. EGNN yields a lightweight, data-efficient model that outperforms state-of-the-art approaches with orders of magnitude fewer parameters, even under scarce and highly imbalanced supervision.

- We conduct a comprehensive evaluation on 7 benchmarks spanning both static and time-series graphs, comparing against 14 baselines, including state-of-the-art methods. EGNN consistently outperforms, demonstrating its effectiveness across diverse settings.

## 2. Spectral Energy of Abnormal Graphs

This section introduces the spectral-energy perspective for graph anomaly detection and summarizes our key observations. Section 2.1 presents the GAD problem formulation, defines spectral energy, and reviews the associated spectral

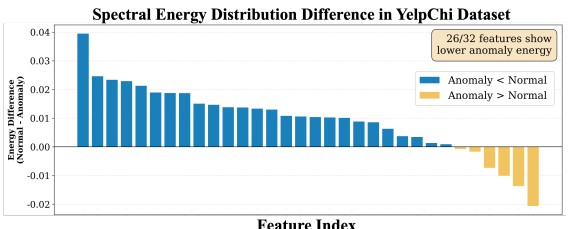

*Figure 1.* Spectral energy distributions in the YelpChi datasets. Each bin shows the average spectral energy difference between normal and anomalous nodes for each feature. Blue indicates anomalous nodes have lower energy, and yellow indicates anomalous nodes have higher energy.

right-shift phenomenon. Section 2.2 identifies an underexplored anomaly pattern in real-world graphs, formalizes it through a precise definition, and validates it on synthetic datasets. Section 2.3 further analyzes how spectral energy manifests at the subgraph level, motivating localized modeling for scalable detection.

### 2.1. Problem Setup and Spectral Energy

**Graph Anomaly Detection.** Let $\mathcal{G} = (\mathcal{V}, \mathcal{E})$ denote an unweighted graph, where $\mathcal{V} = \{v_i\}_{i=1}^N$ is the set of $N$ nodes and $\mathcal{E} \subseteq \mathcal{V} \times \mathcal{V}$ is the edge set. Each node $v_i$ is associated with a $d$ feature vector $x_i \in \mathcal{X} \subseteq \mathbb{R}^d$, and we collect the set of node features $\mathcal{X}$ as $\mathcal{X} = \{x_i\}_{i=1}^N$. Let $\mathbf{x} := [x_1, \ldots, x_N]^\top$, and $\mathbf{x}_f \in \mathbb{R}^N$ denote the column vector of the graph feature indexed $f$. In this work, we focus on *node-level anomalies* and assume the graph structure is reliable, i.e., all edges are anomaly-free. We denote the set of anomalous nodes as $\mathcal{V}_a \subseteq \mathcal{V}$ and the set of normal nodes as $\mathcal{V}_n \subseteq \mathcal{V}$, where $\mathcal{V}_a \cap \mathcal{V}_n = \emptyset$. Accordingly, we formulate the GAD as a binary classification problem. Given the graph $\mathcal{G}$, node features $\mathcal{X}$, and partial node labels $\mathcal{V}_a$ and $\mathcal{V}_n$, the goal is to learn a classifier that assigns labels to each unlabeled node $v \in \mathcal{V}$.

**Spectral Energy Distribution** Let $A \in \mathbb{R}^{N \times N}$ be the adjacency matrix of the graph $\mathcal{G}$, and let $D \in \mathbb{R}^{N \times N}$ denote the associated degree matrix, with diagonal entries $D_{ii} = d_i$, where $d_i = \sum_j A_{ij}$. The normalized Laplacian matrix is defined as $L = I - D^{-1/2}AD^{-1/2}$, where $I$ is the identity matrix of appropriate dimension. When the underlying graph is undirected, i.e., $\mathcal{E}_{ij} = \mathcal{E}_{ji}$, the Laplacian $L$ is symmetric, i.e., $L = L^T$. This symmetry guarantees that $L$ admits real eigenvalues and an orthonormal eigenbasis. Let $L = U\Lambda U^\top$ be the eigendecomposition of $L$, where the eigenvalues satisfy $0 = \lambda_1 \leq \cdots \leq \lambda_N$, and $U = [u_1, u_2, \ldots, u_N]$ is the corresponding orthonormal eigenvectors. Consider the graph attribute $\mathbf{x}_f$ at feature index $f$, its graph Fourier transform is defined as $\hat{\mathbf{x}}_f = U^\top \mathbf{x}_f$. The spectral energy at frequency $\lambda_k (1 \leq k \leq N)$ is then

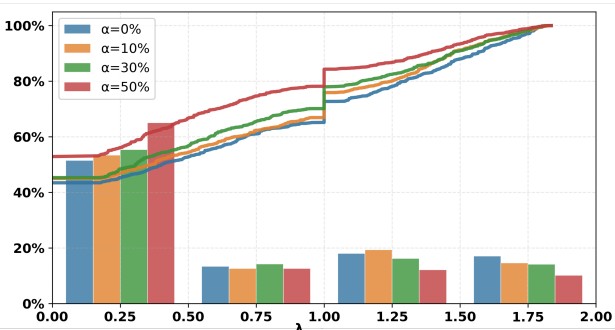

*Figure 2.* Spectral energy distribution on a BA graph under varying fractions ($\alpha$) of neighbor-averaged anomalies. Higher $\alpha$ shifts energy toward lower frequencies, indicating a left-shift phenomenon.

defined as $\frac{\hat{\mathbf{x}}_{k,f}^2}{\sum_{i=1}^N \hat{\mathbf{x}}_{i,f}^2}$.

**Right-Shift Phenomenon**  Previous study (Tang et al., 2022) has shown that anomalous patterns tend to concentrate in high-frequency components in the spectral domain. A right-shift phenomenon is observed when the spectral energy distribution shifts toward higher frequencies Specifically, it characterizes the energy distribution of a graph signal at feature $f$ in the Rayleigh quotient format

$$E_f^{(\text{R})} = \frac{\sum_{k=1}^N \lambda_k \hat{x}_{k,f}^2}{\sum_{k=1}^N \hat{x}_{k,f}^2} = \frac{\mathbf{x}_f^\top L \mathbf{x}_f}{\mathbf{x}_f^\top \mathbf{x}_f}. \quad (1)$$

### 2.2. Camouflage Anomalies

Consider a perfectly smooth, anomaly-free graph signal in which node features are independently and identically distributed as $\mathbf{x}_f \sim \mathcal{N}(\mu\mathbf{1}, \sigma^2 I)$, where $\mathbf{1}$ is an all-ones vector, $\mu$ and $\sigma^2$ stand for mean and variance, respectively. Increasing the proportion of right-shift anomalies enlarges the ratio of $\sigma/|\mu|$. However, not all real-world anomalies exhibit high variance. Previous studies (Hooi et al., 2016; Dou et al., 2020) identify a distinct type of anomaly known as camouflage anomalies, where malicious nodes corrupt the graph by intentionally mimicking the behavior of normal, benign nodes. This anomaly type fundamentally differs from the common high-variance assumption, as such anomalous data are not out-of-distribution. Instead, these nodes closely resemble normal ones, and their insertion into the graph reduces feature variance. A representative example of this anomaly arises in Amazon fake review detection, where paid reviewers tend to produce uniformly high ratings. In this case, the original rating distribution (which naturally contains diverse opinions) is contaminated by an excessive number of identical high-score reviews, causing the overall distribution to concentrate toward a single value and exhibit reduced variance and diversity.

From a probabilistic perspective, this effect can be inter-

preted as a *variance-suppressing contamination process*: injecting identical or highly concentrated samples reduces the empirical variance. In the spectral domain, such a distribution collapse suppresses high-frequency variations and shifts spectral energy toward low-frequency components, producing a *left-shift* phenomenon.

**Definition 2.1** (Left-Shift). Anomalies induced by distribution collapse shift spectral energy from high to low frequencies, producing a leftward shift in the spectral energy distribution, i.e., $\sigma/|\mu|$ decreasing.

**Camouflage anomalies in real-world datasets.**  We further validate the presence of camouflage anomalies on two widely used graph anomaly detection benchmarks, YelpChi and Amazon. Dataset statistics are summarized in Table 1. The results are shown in Fig. 1, with additional results on Amazon provided in the Appendix C. For each dataset, we compute the average local spectral energy of normal and anomalous nodes along each feature dimension. On YelpChi, anomalous nodes exhibit lower spectral energy in half of the features, suggesting that this pattern is not incidental but closely related to distributional collapse. A plausible real-world explanation is that camouflage anomalies may arise from identical or highly concentrated fake reviews, which make anomalous nodes locally smoother and shift their energy toward lower frequencies.

**Left-Shift Phenomenon Validation**  To illustrate the left-shift behavior of camouflage anomalies, we conduct a synthetic experiment on a Barabási–Albert (BA) graph with 1000 nodes. We generate one-dimensional node features from $\mathcal{N}(1, 1)$ for all nodes. Camouflage anomalies are then injected by replacing the features of selected nodes with the average feature value of their immediate neighbors, thereby forcing anomalous nodes to mimic benign local patterns. We vary the anomaly ratio as $\alpha \in \{0\%, 10\%, 30\%, 50\%\}$ to examine its effect on the spectral energy distribution. As shown in Fig. 2, the colored histograms report the proportion of spectral energy within four equal-width eigenvalue intervals, while the curves show the cumulative spectral energy as a function of the eigenvalue $\lambda$. As $\alpha$ increases, spectral energy progressively shifts toward lower frequencies. This confirms that camouflage anomalies increase local smoothness and exhibit the behavior defined in Definition 2.1.

### 2.3. Local Energy Ratio Approximation
Following the experimental setting of the previous work (Tang et al., 2022), we conduct controlled experiments on synthetic graphs with injected anomalies to validate the spectral-energy shift phenomenon using local energy. We consider two graph types: a BA graph with 500 nodes and the Minnesota road network with 2,642 nodes. Normal node features are drawn from $\mathcal{N}(1, 1)$, while anomalous

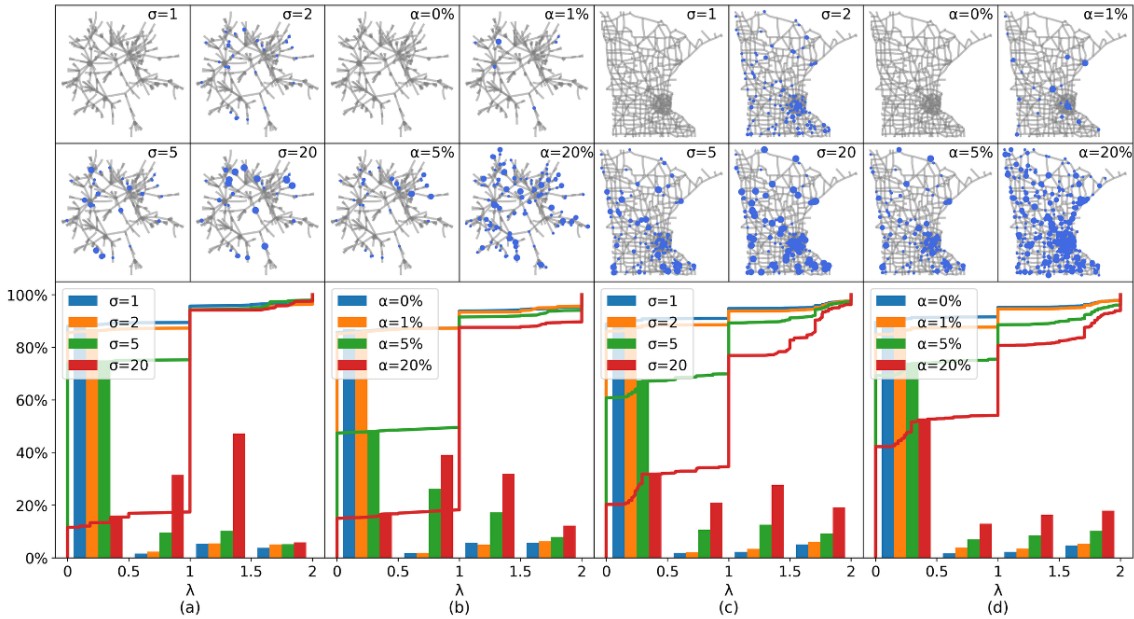

*Figure 3.* Local spectral anomalies under varying anomaly variance $\sigma$ (a, c) and anomaly ratio $\alpha$ (b, d). Plots (a,b) correspond to the BA graph, and plots (c,d) correspond to the Minnesota network. In all plots, higher prevalence and degree of anomalies shift the curves to the right, showing the effectiveness of local spectral energy.

nodes are drawn from $\mathcal{N}(1, \sigma^2)$ with $\sigma > 1$. We examine two anomaly settings: (i) fixing the anomaly ratio to 5% while varying anomaly magnitude $\sigma \in \{1, 2, 5, 20\}$, and (ii) fixing $\sigma = 5$ while varying anomaly ratio $\alpha \in \{0\%, 1\%, 5\%, 20\%\}$.

Our visualization results are shown in Fig. 3. We analyze node-level anomalies and the corresponding spectral energy distributions. When no anomalies are present, more than 80% of the spectral energy is concentrated in the low-frequency region. As the proportion of anomalous nodes increases, spectral energy progressively shifts toward higher frequencies. Specifically, with 1% anomalous nodes, approximately 10% of the low-frequency energy shifts to higher frequencies; with 5% anomalies, more than 20% shifts; and with 20% anomalies, over half of the low-frequency energy moves to the high-frequency region.

The effect of anomaly magnitude is more pronounced: when increasing the anomaly variance from 1 to 20, more than 60% of low-frequency energy is transferred to higher frequencies. As more anomalous nodes with greater magnitudes are introduced, the spectral energy distribution consistently shifts toward higher frequencies, consistent with the global spectral energy distribution. This observation indicates that local spectral energy effectively captures anomalous behavior. The resulting energy curves appear more discrete than their global counterparts, as expected from node-level computation.

## 3. Methodology

In this work, we propose the Energy Graph Neural Network (EGNN), which integrates energy-aware feature transformation with message passing. Specifically, EGNN computes local spectral energy statically, followed by a learnable gating mechanism that adaptively models both left-shift and right-shift energy patterns. The normalized energy is used in the latter graph representation learning framework to capture both normal and anomalous behaviors.

The local energy formulation is introduced in Section 3.1, followed by the energy transformation module in Section 3.2. Section 3.3 presents the proposed left-shift spectral energy formulation, and Section 3.4 details the gating mechanism for adaptive energy modeling. The overall EGNN architecture is described in Section 3.5, and EGNN under the temporal setting is discussed in Section 3.6.

### 3.1. Local Spectral Energy Formulation

As discussed in Section 2.2, anomalous behaviors can be more distinguishable when characterized by local spectral energy ratios. Building on this insight, we propose a localized spectral energy formulation that aligns with the message-passing paradigm of GNNs.

For each node $v_i$, let $\mathcal{N}_i$ of node $\mathcal{N}_i$ denote its 1-hop neighborhood (including $i$). We define the feature vector of the $\mathcal{N}_i$ subgraph as $\mathbf{x}_{\mathcal{N}_i} \in \mathbb{R}^{|\mathcal{N}_i| \times d}$. We then define the local spectral energy for one feature indexed $f$ at the neighbor-

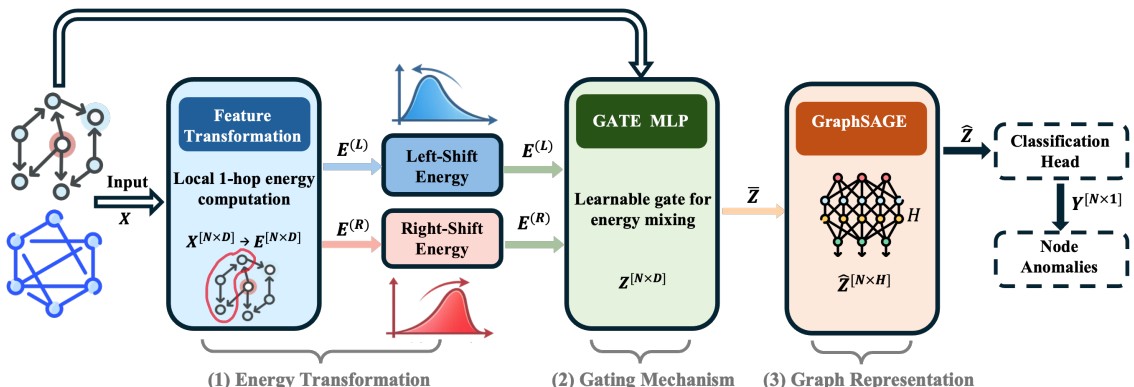

*Figure 4.* Overall framework of EGNN. Raw node features are transformed into two types of spectral energy representations, which are adaptively combined by a learnable gating mechanism. The resulting embeddings are batch-normalized and passed to a graph representation learner followed by a classifier.

hood of node $i$ as the Rayleigh quotient as

$$E_{\mathcal{N}_i,f}^{(R)} = \frac{\mathbf{x}_{\mathcal{N}_i,f}^{\top} L_{\mathcal{N}_i} \mathbf{x}_{\mathcal{N}_i,f}}{\mathbf{x}_{\mathcal{N}_i,f}^{\top} \mathbf{x}_{\mathcal{N}_i,f}}, \qquad (2)$$

where $L_{\mathcal{N}_i}$ denotes the (normalized) Laplacian of the induced subgraph on $\mathcal{N}_i$. When the neighborhood expands to cover the entire graph, i.e., $\mathcal{N}_i = \mathcal{V}$, Eq. (2) reduces to the global spectral energy.

### 3.2. Energy Transformation

The local spectral energy of node $i$ is defined in (2). Although this formulation is localized and enables node-wise energy estimation, it still incurs a high computational cost when applied to large graphs. In the literature, several approaches have been proposed to reduce the cost of computing the quadratic form. Early work employed Chebyshev polynomial approximations to achieve scalable spectral filtering without explicit eigen-decomposition (Defferrard et al., 2016), while later studies leveraged trace-based estimators to approximate this quantity efficiently (Kipf, 2016).

However, these methods rely on approximation, which inevitably introduces discrepancies from the true spectral energy. The goal of energy feature transformation is to design an energy transformation that maps node features from $\mathbf{x}$ to energy representations $E \in \mathbb{R}^{N \times d}$ without losing structural information. To this end, we propose a feature-level energy transformation based on localized spectral energy. The normalized local spectral energy of node $v_i$ at feature $a$ is formulated cumulatively as follows

$$E_{i,f}^{(R)} = \frac{\sum_{j \in \mathcal{N}(i)} w_{ij} \left( \frac{x_i}{\sqrt{d_i}} - \frac{x_j}{\sqrt{d_j}} \right)^2}{\left( \frac{x_i}{\sqrt{d_i}} \right)^2 + \sum_{j \in \mathcal{N}(i)} \left( \frac{x_j}{\sqrt{d_j}} \right)^2} \qquad (3)$$

where $w_{ij}$ denotes the edge weight and $x_j^f$ represents the feature value of node $j$ (a neighbor of node $i$) at feature

dimension $f$. We normalize the formulation by the node degree $\sqrt{d_i}$ to account for varying neighborhood sizes. We refer the reader to Appendix A.2 for the detailed derivation.

In general, edge weights do not necessarily fall within linear relationships (e.g., pipe connections in water distribution networks). In such cases, $w_{ij}$ can be parameterized by a learnable function to model edge relationships. In this paper, however, we focus on the unweighted setting where all edges are treated equally, i.e., $w_{ij} = 1$, and leave the weighted case for future work.

### 3.3. Left-Shift Spectral Energy

As discussed earlier, the emergence of the left-shift in the spectral energy distribution is not coincidental, but rather a consequence of distribution collapse. The original spectral energy in Eq. (1) is defined as the ratio between the high-frequency energy $\sum_{k=1}^{N} \lambda_k \hat{x}_k^2$ and the total signal energy $\sum_{k=1}^{N} \hat{x}_k^2$. In practice, high-frequency components are dominated by the largest eigenvalues and primarily capture highly variant, non-smooth anomalies. However, anomalies that reside in the low-frequency domain cannot be effectively detected by this formulation.

To capture low-frequency spectral energy while preserving the Rayleigh quotient form, we construct a flipped Laplacian by reversing the eigenvalue spectrum without permuting the eigenbasis. When using the normalized graph Laplacian, whose eigenvalues are bounded in $[0, 2]$, the flipped Laplacian can be expressed as $2I - L$. The corresponding left-shift spectral energy is then given by

$$E_{\mathcal{N}_i,f}^{(L)} = \frac{\mathbf{x}_{\mathcal{N}_i,f}^{\top}(2I - L_{\mathcal{N}_i})\mathbf{x}_{\mathcal{N}_i,f}}{\mathbf{x}_{\mathcal{N}_i,f}^{\top} \mathbf{x}_{\mathcal{N}_i,f}} = 2 - E_{\mathcal{N}_i,f}^{(R)} \quad (4)$$

where $R$ denotes the original spectral energy defined in Eq. (1). And the normalized left-shift local spectral energy of node $v_i$ at feature $f$ is expressed as $E_{\mathcal{N}_i,f}^{(L)} = 2 - E_{\mathcal{N}_i,f}^{(R)}$.

*Table 1.* Statistics of the static and time-series datasets.

| Static graph datasets | | | | |
|---|---|---|---|---|
| **Dataset** | **Nodes** | **Edges** | **Anom.** | **Feat.** |
| Amazon | 11,944 | 4,398,392 | 6.87% | 25 |
| YelpChi | 45,954 | 3,846,979 | 14.53% | 32 |
| T-Finance | 39,357 | 21,222,543 | 4.58% | 10 |
| T-Social | 5,781,065 | 73,105,508 | 3.01% | 10 |
| Time-series datasets | | | | |
| **Dataset** | **Domain** | **Chan.** | **Samples** | **Anom.** |
| SWaT | Water | 51 | 946,719 | 11.98% |
| WADI | Water | 127 | 172,738 | 5.99% |
| MSL | Space | 55 | 132,046 | 10.72% |

### 3.4. Gating Mechanism

Synthetic benchmarks (e.g., Barabási–Albert and Minnesota road networks) are often generated by injecting high-variance perturbations, and such anomalies typically exhibit high variance and present a right-shift in the spectral domain. In our experiments on the Amazon dataset, we observe that a portion of anomalies also exhibit low spectral energy. Therefore, relying on only right-shift or left-shift energy is insufficient to capture all anomaly types; instead, their combination is necessary to distinguish anomalous nodes from normal ones in terms of spectral energy. To achieve this, we introduce an adaptive gating mechanism that dynamically learn the combination of right-shift and left-shift energy representations.

Specifically, we design a lightweight MLP as a feature-wise gating map from the raw node attributes $\mathbf{x}$ to a binary weight matrix $G \in \mathbb{B}^{N \times d}$ given as $G = \mathrm{MLP}(\mathbf{x})$ that controls the combination of the right- ($E_f^{(R)}$) and the left-shift energy ($E_f^{(L)}$) representations. The fused energy embedding is computed as $\mathbf{Z} = G \odot E_f^{(L)} + (1 - G) \odot E_f^{(R)}$ where $\odot$ denotes element-wise multiplication. Finally, we apply batch normalization to get $\bar{\mathbf{Z}}$, feeding it into the GNN backbone for anomaly classification.

### 3.5. Energy Driven Graph Neural Network

The Energy-Driven Graph Neural Network (EGNN) builds upon the localized spectral energy transformation and the adaptive gating mechanism. The overall architecture of the proposed method is illustrated in Fig. 4. Given the fused energy representation $\bar{\mathbf{Z}}$, EGNN applies a GraphSAGE backbone to propagate energy-aware information over the graph $\mathbf{H} = \mathrm{AGG}(\bar{\mathbf{Z}}_0, \bar{\mathbf{Z}}_1, \cdots \bar{\mathbf{Z}}_n)$. The resulting node embeddings $\mathbf{H}$ are then fed into a linear classifier to compute the anomaly probability $p_i \in (0, 1)$ for each node.

**Learning Objective** Due to the inherent class imbalance in anomaly detection, we train EGNN using a weighted cross-entropy loss:

$$\mathcal{L} = - \sum_{i \in \mathcal{V}_{\mathrm{train}}} \left[ \alpha\, y_i \log(p_i) + (1 - y_i) \log(1 - p_i) \right], \quad (5)$$

where $y_i \in \{0, 1\}$ is the ground-truth label and $\alpha$ is the anomaly ratio. The parameters of the gating MLP ($\Theta_{\mathrm{gate}}$), the GraphSAGE backbone ($\Theta_{\mathrm{conv}}$), and the classifier ($\Theta_{\mathrm{cls}}$) are jointly optimized.

### 3.6. Time-Series Graph Extension

Given a temporal graph sequence $\mathcal{S} = \{\mathcal{G}_1, \mathcal{G}_2, \ldots, \mathcal{G}_L\}$, we define each graph at time $t$ as $\mathcal{G}_t = (\mathcal{V}, \mathcal{E}_t, \mathbf{X}_t)$, where $\mathcal{V}$ denotes a fixed set of $N$ nodes, $\mathcal{E}_t$ represents the set of edges at time $t$, and $\mathbf{X}_t \in \mathbb{R}^{N \times d}$ is the corresponding node feature matrix.

**Unified EGNN for Time-Series Graphs.** We extend EGNN to time-series graph anomaly detection using the same unified architecture as in the static setting. Instead of computing feature-wise energy on a single snapshot (Eq. 4), we compute the localized spectral energy at each time step $t$ within a sliding window $T$ (of length $w$), and apply energy transformation and gating modules across all time steps.

**Robustness to long windows.** The window length $w$ is a tunable hyperparameter in temporal anomaly detection, and many sequence models suffer from degraded performance under large windows due to optimization instability (e.g., vanishing/exploding gradients). In contrast, EGNN does not rely on recurrent or attention-based temporal encoders (e.g., RNNs or Transformers). Temporal dependencies are captured implicitly through energy-aware representations and the gating mechanism, while GraphSAGE performs spatial aggregation on each snapshot. As a result, increasing $w$ does not introduce deep temporal backpropagation paths, and longer windows can even provide richer context for improving node representations.

**Data-efficient modeling.** EGNN is lightweight: the gating module is implemented as a shallow MLP and the backbone uses only a few graph convolution layers. Compared to parameter-heavy temporal architectures, EGNN contains substantially fewer learnable parameters, reducing overfitting risk and enabling more data-efficient training in real-world settings where labeled anomalies are scarce.

## 4. Experiments

In this section, we compare EGNN with state-of-the-art approaches across both temporal and static settings. We evaluate detection accuracy, assess scalability on the large-scale T-Social dataset, and examine the effectiveness of

*Table 2.* Performance on static graph datasets. Runtime reports the execution time for a single run.

| Model | Amazon | | | YelpChi | | | T-Finance | | | T-Social | | | |
|---|---|---|---|---|---|---|---|---|---|---|---|---|---|
| | F1-m | AUC | PRC | F1-m | AUC | PRC | F1-m | AUC | PRC | F1-m | AUC | PRC | Time(s) |
| GCN | 63.22 | 82.20 | 34.15 | 55.70 | 58.91 | 22.42 | 72.63 | 85.59 | 43.45 | 65.34 | 78.80 | 23.96 | 2041 |
| ChebyNet | 91.03 | 95.76 | 86.18 | 71.59 | 84.30 | 54.46 | 85.70 | 93.29 | 75.37 | 58.31 | 73.56 | 12.04 | 1380 |
| GAT | 85.70 | 93.29 | 75.37 | 65.92 | 78.33 | 41.34 | 81.56 | 91.69 | 55.90 | 67.08 | 85.47 | 25.65 | 2945 |
| GIN | 88.93 | 92.32 | 80.29 | 65.20 | 76.59 | 39.51 | 79.65 | 84.41 | 51.83 | 52.61 | 66.37 | 5.92 | 3611 |
| GraphSAGE | 75.28 | 88.90 | 66.20 | 68.59 | 82.17 | 45.80 | 57.04 | 57.98 | 8.71 | 58.42 | 74.04 | 9.44 | 3063 |
| SGC | 63.06 | 77.10 | 23.68 | 51.35 | 51.91 | 15.76 | 57.92 | 51.91 | 15.76 | 42.49 | 48.93 | 3.04 | 2109 |
| GT | 89.14 | 90.75 | 76.20 | 67.27 | 79.86 | 44.89 | 64.63 | 78.50 | 19.87 | 63.66 | 83.43 | 20.82 | 2296 |
| BernNet | 91.31 | 93.83 | 84.01 | 69.19 | 82.14 | 48.89 | 81.48 | 90.66 | 52.90 | 51.94 | 64.68 | 4.88 | 1945 |
| PC-GNN | 67.04 | 81.80 | 29.36 | 56.30 | 59.55 | 22.91 | 85.62 | 92.17 | 73.70 | 51.19 | 72.88 | 13.77 | 15399 |
| BWGNN | 91.40 | 96.17 | 86.42 | 71.78 | 84.33 | 54.67 | 84.67 | 93.12 | 73.84 | 81.78 | 94.32 | 60.81 | 3045 |
| UniGAD | 90.46 | **96.60** | 86.65 | 71.23 | 83.69 | 53.97 | 89.34 | 95.14 | **84.71** | 78.67 | 91.65 | 58.33 | 3981 |
| **Ours** | **91.52** | 96.32 | **89.40** | **76.89** | **88.10** | **66.26** | **89.60** | **95.39** | 84.39 | **95.40** | **99.69** | **95.89** | 3712 |

capturing left-shift anomalies by contrasting models trained only on right-shift energy with those trained on both.

**Datasets** To evaluate, we conduct experiments on seven public benchmarks, including four static graphs and three time-series graphs. Table 1 summarizes their statistics. **Amazon** (McAuley & Leskovec, 2013) and **YelpChi** (Rayana & Akoglu, 2015) target fake-review detection and exhibit mixed left-shift and right-shift spectral patterns, suggesting the coexistence of distribution collapse and outliers. **T-Finance** (Tang et al., 2022) focuses on anomalous account detection in transaction networks, while **T-Social** (Tang et al., 2022) considers large-scale social graphs. Notably, T-Social contains over 5M nodes and 73M edges, and is included to demonstrate EGNN's scalability.

The Secure Water Treatment (**SWaT**) dataset (Mathur & Tippenhauer, 2016) is collected from a real-world water treatment testbed and serves as a representative cyber-physical system benchmark. The Water Distribution (**WADI**) dataset (Ahmed et al., 2017) extends SWaT to a larger-scale water distribution network with more complex pipeline structures. The **MSL** dataset (Hundman et al., 2018) consists of spacecraft telemetry, where anomalies correspond to system faults and unexpected operational behaviors.

**Experimental Setting** To ensure a fair comparison, all baseline methods are trained and evaluated under identical experimental settings, including the same number of training epochs, learning rate, and train/validation/test splits. We apply a 40%/20%/40% split for training, validation, and testing across all four datasets. Each method is run 10 times on Amazon, Yelp, and T-Finance, and we report the mean and standard deviation of the results. For T-Social, due to its large scale and high computational cost, we run each method 5 times and report the mean and standard deviation. For time-series datasets, all methods are evaluated over 10 runs under the standard train-on-normal and test-

on-mixed protocol for the unsupervised baseline. To enable semi-supervised training, we additionally sample a fixed 1% labeled subset from the mixed data. No point adjustment is applied in any evaluation. Additional details are provided in Appendix D.

**Baseline Methods** In the static graph setting, we compare the proposed EGNN with a diverse set of representative graph learning models, covering both classical and state-of-the-art graph neural network architectures, ranging from spectral-based to attention-based methods. Specifically, we include the following baseline models: **GCN** (Kipf, 2016), **ChebyNet** (Defferrard et al., 2016), **GAT** (Veličković et al., 2017), **GIN** (Xu et al., 2018), **GraphSAGE** (Hamilton et al., 2017), **SGC** (Wu et al., 2019), **GT** (Dwivedi & Bresson, 2020), **BernNet** (He et al., 2021), **PC-GNN** (Liu et al., 2021), and **BWGNN** (Tang et al., 2022). For time-series graphs, we include **GDN** (Deng & Hooi, 2021), **TranAD** (Tuli et al., 2022), and **USAD** (Audibert et al., 2020) as representative baselines. For detailed descriptions of these baseline models, we refer the reader to Appendix B.2.

**Metrics** To comprehensively evaluate model performance, we adopt three widely used metrics: **Macro-F1**, Area Under the Receiver Operating Characteristic Curve (**AUROC**), and Area Under the Precision-Recall Curve (**AUPRC**), where AUPRC is computed using average precision. Additional details are provided in the appendix B.3.

### 4.1. Experimental Results

**Performance in Static Graph** *EGNN achieves over 95% accuracy in the most challenging dataset and outperforms all the baselines.* The performance of the proposed method and the baselines is reported in Table 2. On the small-scale dataset(Amazon, YelpChi, and T-Finance), strong baselines such as BWGNN, BernNet, and the classical ChebyNet already achieve over 90% F1-macro. Although the improve-

*Table 3.* Performance comparison on time-series datasets.

| Model | MSL | | | SWaT | | | WADI | | | Avg. F1 | #Params |
|---|---|---|---|---|---|---|---|---|---|---|---|
| | AUC | PRC | F1-m | AUC | PRC | F1-m | AUC | PRC | F1-m | | |
| TranAD | 73.57 | 92.71 | 68.14 | 81.49 | 70.62 | 76.45 | 43.53 | 5.24 | 48.50 | 64.36 | 261,243 |
| USAD | 80.60 | 95.02 | 63.71 | 78.21 | 69.43 | 71.52 | 49.87 | 5.63 | 48.35 | 61.19 | 1.28M |
| GDN | 65.79 | 88.80 | 57.56 | 80.57 | **71.47** | 75.45 | 46.99 | 4.96 | 48.18 | 60.39 | 5,121 |
| **EGNN** | **86.63** | **97.28** | **70.50** | **93.70** | 70.11 | **86.03** | **67.88** | **20.27** | **61.41** | **72.65** | 53,953 |

ment on Amazon is relatively marginal due to the simplicity of the dataset, EGNN significantly outperforms all baselines on more challenging and large-scale datasets, including YelpChi, T-Finance, and T-Social. In particular, EGNN improves F1-macro by more than 16% on the T-Social dataset. This substantial gain highlights the scalability of EGNN on large graphs. Moreover, EGNN consistently achieves superior AUPRC, even on the largest and most challenging dataset, achieving over 95%, while none of the baseline methods reach comparable performance in AUPRC. This further demonstrates EGNN's strong capability in identifying anomalous nodes, especially on large-scale datasets.

**Performance in Time-Series Graph** *EGNN achieves the highest average F1 score among three datasets and with fewer parameters.* Table 3 reports the performance of different methods on three benchmark time-series anomaly detection datasets. EGNN consistently outperforms deep all method across all three datasets in terms of F1-macro. On the challenging SWaT dataset, although GDN achieves the highest AUPRC, EGNN attains an F1-score of 86.03, significantly outperforming TranAD (76.45) and USAD (71.52). Similarly, on WADI, EGNN improves the F1-score to 61.41, surpassing all neural baselines by a large margin. Moreover, EGNN uses substantially fewer parameters than TranAD and USAD, while GDN has only a small number of parameters due to the lack of temporal learning, which partly explains its inferior performance. These results highlight the effectiveness and efficiency of EGNN in capturing temporal anomaly patterns under weakly supervised settings.

### 4.2. Sensitivity Analysis

**Effectiveness of Left-Shift Spectral Energy** We provide ablation results in Table 4, comparing the full model with variants that use only a single energy branch, fixed equal-weight fusion, and a semantically irrelevant simple random-walk feature (SRW). The results show that each component contributes to the final performance. (1) Neither spectral branch alone is sufficient, as Amazon and YelpChi exhibit mixed spectral patterns that require both left-shift and right-shift information. (2) Learnable fusion is necessary because fixed equal-weight fusion cannot adapt to feature-wise spectral differences and leads to degraded performance. (3) The gain from left-shift energy is not simply due to increased

feature dimensionality, since replacing it with SRW also reduces performance. These results confirm that left-shift spectral energy provides meaningful information.

*Table 4.* Ablation results of the effectiveness of left-shift spectral energy, different backbones, and gate designs on Amazon and YelpChi datasets.

| Setting | Amazon | | | YelpChi | | |
|---|---|---|---|---|---|---|
| | F1-m | AUC | PRC | F1-m | AUC | PRC |
| $E^{(R)}$ only | 87.20 | 94.97 | 81.36 | 72.21 | 83.61 | 56.19 |
| $E^{(L)}$ only | 87.73 | 95.08 | 83.50 | 71.77 | 83.47 | 55.69 |
| Fixed | 72.86 | 86.27 | 45.72 | 49.44 | 50.22 | 14.78 |
| SRW | 74.23 | 88.04 | 50.75 | 50.31 | 50.67 | 14.97 |
| With GCN | 75.40 | 87.84 | 49.68 | 58.13 | 63.68 | 27.44 |
| With GAT | 60.03 | 62.76 | 19.88 | 62.55 | 73.33 | 35.25 |
| CA | 89.35 | 95.16 | 84.55 | 75.19 | 86.49 | 62.34 |
| Bilinear | 88.78 | 93.62 | 82.24 | 72.80 | 84.74 | 57.84 |
| Original | **91.52** | **96.32** | **89.40** | **76.89** | **88.10** | **66.26** |

**Component Analysis** We further evaluate the proposed method with different GNN backbones and gating mechanisms. Table 4 also compares different GNN backbones. Replacing GraphSAGE with GCN or GAT consistently reduces performance. We attribute this to GraphSAGE having a better ability to preserve self-information during neighbor aggregation. In contrast, GCN introduces additional spectral-style propagation that may blur the intended spectral semantics, while GAT applies attention-based reweighting that appears unnecessary in this setting. As shown in Table 4, replacing the MLP gate with cross-attention (CA) or bilinear fusion consistently degrades performance on both datasets, suggesting that simple feature-wise gating is sufficient to balance left- and right-shift spectral energies. CA is likely redundant because local spectral energy already encodes neighborhood information, while bilinear fusion increases model complexity without improving performance.

**Parameter Analysis** We conduct a parameter sensitivity analysis to examine the effect of several key hyperparameters on EGNN. Figure 5 shows the performance of EGNN under different temporal window sizes on time-series datasets and different hop numbers on static graph datasets. The results indicate that varying the temporal window size

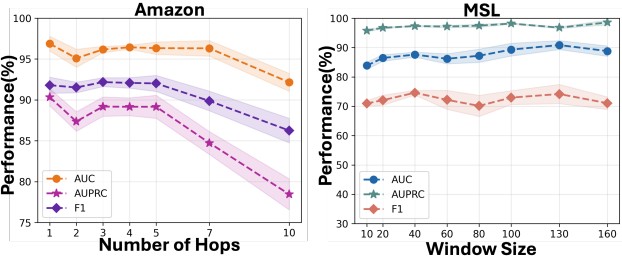

*Figure 5.* Performance of EGNN on the Amazon dataset under different hop numbers and on the MSL dataset under different window sizes.

has only a limited impact on performance; even relatively small ($w = 10$) and large ($w = 160$) values do not cause noticeable degradation, suggesting that EGNN is robust to this parameter. As the hop size increases, local spectral energy becomes less discriminative at the node level. In the extreme case where the hop size approaches the graph diameter, each node recovers nearly the same global spectral energy. Although 3-hop achieves the best F1 score, its improvement over 1-hop is marginal while introducing a higher computational cost. We refer readers to Appendix B.7 for detailed results on the effect of hop size. Therefore, 1-hop provides a better trade-off between effectiveness and efficiency.

**Complexity Analysis**  The energy computation in Eq. 3 costs $O(|\mathcal{E}|F)$ and is performed once as preprocessing. The gating module adds $O(NFGL_g)$ complexity. Each GraphSAGE layer costs $O(|\mathcal{E}|H + NH^2)$. Therefore, the per-epoch training complexity is $O(|\mathcal{E}|H + NH^2 + NFGL_g)$. Since real-world graphs typically satisfy $|\mathcal{E}| \gg N$, the sparse message-passing term dominates in practice, yielding an overall complexity of approximately $O(|\mathcal{E}|H)$. We further report empirical computational costs in Appendix B.8. Although our method introduces additional spectral-energy and gating computations, it remains scalable and consistently improves detection performance across datasets.

## 5. Related Work

**Static Graph Anomaly Detection.**  Graph Anomaly Detection (GAD) on static graphs aims to identify abnormal nodes, edges, or subgraphs based on structural and attribute information. Early methods primarily relied on graph representation learning with shallow models, such as random walk embeddings (Perozzi et al., 2014) or matrix factorization (Grover & Leskovec, 2016). With the development of Graph Neural Networks (GNNs), a large body of work has adopted message passing architectures for anomaly detection. Representative approaches include GCN- (Kipf, 2016), GAT- (Veličković et al., 2017), and GraphSAGE- (Hamilton et al., 2017) based methods, which learn node representa-

tions by aggregating neighborhood information. Several recent studies have explored spectral and frequency-based perspectives to characterize anomalies. For example, BWGNN (Tang et al., 2022) and BernNet (He et al., 2021) use graph spectral filters to model high-frequency components, motivated by the observation that anomalies tend to exhibit large feature variations. Other methods, such as PC-GNN (Liu et al., 2021) and subgraph-based approaches, focus on capturing local structural irregularities. However, most existing static GAD methods implicitly assume that anomalies are associated with high-frequency or high-variance patterns, which limits their ability to detect diverse anomaly types.

**Time-Series Graph Anomaly Detection**  Time-Series Anomaly Detection (TSAD) is challenging due to anomalies often arising from complex temporal dependencies and inter-variable correlations. Sequence-centric models, such as Anomaly Transformer (Xu et al., 2021) and TranAD (Tuli et al., 2022), mainly rely on Transformer architectures to model temporal dynamics. These methods are effective for capturing long-range temporal patterns, but they usually treat variables as independent channels or only implicitly model their interactions, thereby overlooking explicit relational dependencies among variables. Recent studies have introduced graph-based formulations for TSAD, where each variable is represented as a node and edges encode inter-variable dependencies. For instance, GDN (Deng & Hooi, 2021) learns a dynamic dependency graph based on variable similarity and achieves competitive performance on standard benchmarks even without complex temporal architectures. However, most existing graph-based anomaly detection methods are still designed for static settings (Ding et al., 2019). When applied to dynamic graphs, where both topology and node attributes evolve, these methods struggle to jointly capture structural evolution and temporal dependencies. As a result, anomaly detection on dynamic graphs remains a challenging and relatively under-explored problem (Ma et al., 2021).

## 6. Conclusion and Future Direction

In this work, we identified camouflaged anomalies, a new class of patterns that induce a left-shift phenomenon in spectral energy distributions. Based on this observation, we proposed EGNN, an energy-aware framework for GAD that broadens the scope of detectable node attribute anomalies. We further extended spectral energy modeling to temporal graphs with a focus on scalability and data efficiency. Our experiments demonstrated that EGNN consistently outperforms state-of-the-art methods with orders of magnitude fewer parameters across diverse settings. Directed graphs and other granularity anomalies, e.g., edge-, subgraph-, and graph-, will be explored in future work. Detailed limitations are further discussed in Appendix D.

## Acknowledgment

This work was supported in part by the National Science Foundation under grants NSF 2443803 and 2230087. Any opinions, findings, and conclusions or recommendations expressed in this material are those of the authors and do not necessarily reflect the views of the National Science Foundation.

## Impact Statement

This paper presents EGNN, a spectral-energy framework for spatio-temporal graph anomaly detection that captures both high-variance and camouflaged low-variance anomalies. Its main benefit is improving the detection of fraud, coordinated manipulation, and abnormal behaviors in networked systems, thereby reducing economic losses and improving operational reliability.

However, deployment risks remain. False positives may unfairly affect legitimate entities, while false negatives may miss critical threats. Since EGNN is semi-supervised, it may also inherit biases from historical labels. Its assumptions on node-attribute anomalies over undirected and unweighted graphs may further limit robustness under distribution shift or in directed and weighted settings. Responsible deployment should include threshold calibration, subgroup-level audits, privacy-preserving practices, and human oversight in high-stakes applications.

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

# A. Notations and Derivations

In this section, we summarize the notations utilized in this paper and present the derivation of (4).

## A.1. Notations

*Table 5.* Notation table.

| Symbol | Meaning | Symbol | Meaning |
|---|---|---|---|
| $\mathcal{G}$ | Graph | $\mathcal{V}$ | Set of nodes |
| $\mathcal{E}$ | Set of edges | $N$ | The number of nodes |
| $A$ | Adjacency matrix | $D$ | Degree matrix |
| $L$ | Laplacian matrix | $U$ | Eigenvector matrix |
| $\Lambda$ | Eigenvalues | $x$ | Node feature vector |
| $\mathcal{X}$ | The set of node feature vectors | $\mathbf{x}$ | Node feature matrix |
| $\hat{\mathbf{x}}_f$ | Fourier transform of $\mathbf{x}$ in feature $f$ | $L_{\mathcal{N}_i}$ | Subgraph Laplacian matrix |
| $E_{i,f}^{(L)}$ | Left-shift spectral energy at node $v_i$ feature $f$ | $E_{i,f}^{(R)}$ | Right-shift spectral energy at node $v_i$ feature $f$ |
| $g$ | Feature-wise weight | $Z$ | Representation after gate mechanism |
| $\mathbf{H}$ | Node embedding | $\mathcal{L}$ | Training loss |
| $\alpha$ | Anomaly ratio | $F$ | Feature dimension |
| $G$ | Hidden dimension of the gating network | $L_g$ | Number of layers in the gating network |
| $H$ | Hidden dimension of the GNN layer | $|\mathcal{E}|$ | The number of edges |

## A.2. Derivation of equation (3).

We have $L_{\mathcal{N}_i} = I - D_{\mathcal{N}_i,f}^{-1/2} A_{\mathcal{N}_i,f} D_{\mathcal{N}_i,f}^{-1/2}$. Substituting $L_{\mathcal{N}_i}$ into (2) for neighborhood $\mathcal{N}_i$ and expanding via matrix multiplication, we obtain $E_{\mathcal{N}_i,f}^{(R)} = E_{num}/E_{den}$, where the numerator and denominator are given by

$$E_{num} = \sum_{j \in \mathcal{N}(i)} w_{ij} \left( \frac{x_{i,f}}{\sqrt{d_i}} - \frac{x_{j,f}}{\sqrt{d_j}} \right)^2, \quad E_{den} = \frac{(x_{i,f})^2}{d_i} + \sum_{j \in \mathcal{N}(i)} \frac{(x_{j,f})^2}{d_j}$$

# B. Additional Experiment Details

## B.1. Experimental Setup

All the experiments were run in a single workstation, which is equipped with an AMD Ryzen Threadripper PRO 7975WX processor (32 cores, 64 threads), 128 GB of DDR5 RAM, and an NVIDIA RTX 6000 Ada Generation GPU with 48 GB of VRAM.

## B.2. Baseline Description

- **GCN**: GCN aggregates neighbor information via graph convolution to update node representations.

- **ChebyNet**: ChebyNet approximates spectral graph convolution with Chebyshev polynomials to capture multi-hop structure efficiently.

- **GAT**: GAT uses attention to assign different weights to neighbors, focusing message passing on the most relevant nodes.

- **GIN**: GIN uses an injective aggregation function and is as expressive as the 1-Weisfeiler–Lehman test in distinguishing graph structures.

- **GraphSAGE**: GraphSAGE learns node embeddings by sampling and aggregating information from local neighborhoods.

- **SGC**: SGC simplifies GCN by removing nonlinearities and collapsing propagation into a single linear model on pre-smoothed features.

- **GT**: GT adapts Transformers to graphs by masking self-attention with graph structure to improve efficiency and inductive bias.

- **BernNet**: BernNet learns adaptive graph filters using a Bernstein polynomial approximation over the normalized Laplacian spectrum.

- **PC-GNN**: PC-GNN addresses class imbalance with label-balanced sampling and learns a distance function to select informative neighbors and suppress noisy links.

- **BWGNN**: BWGNN targets "right-shift" anomalies using Beta wavelet kernels to construct flexible band-pass graph filters for high-frequency signals.

- **UniGAD**: UniGAD is a unified graph anomaly detection framework, which models anomalies across multiple levels (node, edge, and graph) within a single architecture.

- **GDN**: GDN models multivariate time series as a variable graph and scores anomalies by deviations from learned dependency patterns.

- **TranAD**: TranAD is a Transformer-based method that detects anomalies via reconstruction error, enhanced by an adversarial-style training objective.

- **USAD**: USAD is an unsupervised autoencoder method with two decoders trained iteratively to produce robust reconstruction-based anomaly scores.

## B.3. Metrics

**AUPRC (Area Under the Precision-Recall Curve)**    AUPRC is used in a classification model to show the relationship between precision and recall at different threshold levels, which computes the area beneath the Precision-Recall curve.

**AUROC (Area Under the Receiver Operating Characteristic Curve)**    AUROC evaluates a model's ability to distinguish the positive and negative classes by measuring the area under the ROC curve. The ROC curve plots the true positive rate against the false positive rate for varying decision thresholds. The ROC approach to 1 shows it has perfect ability to distinguish the different classes.

**F1-macro**    F1-macro computes the average score over all classes by treating each class with the same weight. This metric is widely used in highly imbalanced classification tasks, such as graph anomaly detection.

## B.4. Details of Training in static graph

All models are trained for a maximum of 200 epochs using the Adam optimizer with a learning rate of 0.01 and a weight decay of $5 \times 10^{-4}$. Early stopping is applied with a patience of 50 epochs based on validation performance. Each experiment is repeated 10 times with different random seeds. The dataset is split into training, validation, and test sets with ratios of 0.4, 0.2, and 0.4, respectively. We use the model provided in GADBench, and Table 6 shows the hyperparameters we used for all baselines.

*Table 6.* Baseline models and hyperparameters.

| Model | Key Parameters |
|---|---|
| GCN | h_feats=32, num_layers=1, drop_rate=0.15 |
| SGC | h_feats=32, k=2, drop_rate=0.15 |
| GIN | h_feats=32, num_layers=2, drop_rate=0.15 |
| GraphSAGE | h_feats=32, num_layers=2, drop_rate=0.15 |
| GAT | h_feats=32, num_layers=2, num_heads=4, drop_rate=0.15 |
| GT | h_feats=32, num_layers=2, num_heads=4, drop_rate=0.15 |
| BWGNN | h_feats=64, num_layers=2, drop_rate=0.00 |
| BernNet | h_feats=32, orders=3, drop_rate=0.15 |
| PCGNN | h_feats=64, num_layers=2, del_ratio=0.4, add_ratio=0.4 |
| ChebNet | h_feats=32, k=2, drop_rate=0.15 |
| UniGAD | h_feats=32, num_layers=2, encoder=bwgnn, epoch_pretrain=50, mask_ratio=0.5 |

*Table 7.* Ablation study on different hop sizes.

| Setting | Amazon | | | YelpChi | | |
|---|---|---|---|---|---|---|
| | F1-m | AUC | PRC | F1-m | AUC | PRC |
| 1-hop | 91.80 | 96.88 | 90.33 | 76.91 | 88.12 | 66.10 |
| 3-hop only | **92.18** | 96.17 | **89.18** | **78.21** | **88.74** | **67.41** |
| 5-hop only | 92.02 | **96.33** | 89.16 | 77.35 | 88.11 | 66.82 |
| 7-hop only | 89.97 | 96.31 | 84.73 | 74.36 | 87.84 | 63.07 |

## B.5. Details of Training in time-series graph

All three anomaly detection models (GDN, TranAD, and USAD) are evaluated on the TGAD benchmark datasets, using a consistent experimental setup. The input data is preprocessed using MinMaxScaler normalization, and sliding windows of size 15 are employed to construct temporal sequences from the multivariate time series. During training, GDN uses a stride of 5, while TranAD and USAD use a stride of 1; all models use a stride of 1 during testing. The models are trained for 50–150 epochs (50 for GDN, and 150 for TranAD and USAD) with a learning rate of 0.001 and batch sizes of 128 for GDN and USAD, and 32 for TranAD. A validation split (10–30% of the training data) is used for early stopping and threshold tuning. The optimal classification threshold is selected by sweeping over score percentiles to maximize the Macro F1 score on the validation set. All experiments report seven evaluation metrics: Macro F1, Binary F1, Recall, Precision, AUROC, and AUPRC. Multiple runs with different random seeds are conducted to compute the mean and standard deviation, and reproducibility is ensured by fixing random seeds for Python, NumPy, and PyTorch.

## B.6. Additional Experimental Results

In Table 9, we report the average value and standard deviation of 10 runs on YelpChi, Amazon, T-Finance, and T-social.

## B.7. Number of hops analysis

Table 7 shows that increasing the hop size does not consistently improve performance. While 3-hop achieves the best overall results on YelpChi and competitive performance on Amazon, larger hops, such as 7-hop, degrade performance, suggesting that overly broad neighborhoods may dilute local anomaly signals.

## B.8. Runtime Analysis

Table 8 reports the computational cost of different models for one epoch across four graph anomaly detection datasets. Our method requires higher GFLOPs than lightweight GNN baselines such as GCN and GraphSAGE, but remains more efficient than GAT on Amazon, YelpChi, and T-Finance while providing stronger anomaly detection performance.

*Table 8.* Computation cost measured by GFLOPs.

| Models | Amazon | YelpChi | T-Finance | T-Social |
|---|---|---|---|---|
| GCN | 1.78 | 2.15 | 5.70 | 69.5 |
| GraphSAGE | 6.78 | 9.77 | 28.27 | 646.0 |
| GAT | 67.57 | 61.51 | 298.0 | 1680 |
| BWGNN | 4.62 | 6.82 | 19.31 | 497.9 |
| UniGAD+BWGNN | 5.03 | 8.49 | 20.41 | 659.9 |
| Ours | 18.65 | 30.38 | 57.92 | 1990 |

## C. Left-Shift Spectral Energy in Amazon dataset

This figure shows the spectral energy distribution on the Amazon dataset. Each bin represents the average spectral energy for one feature across all nodes. Among the 25 features, 12 exhibit lower spectral energy for anomalous nodes.

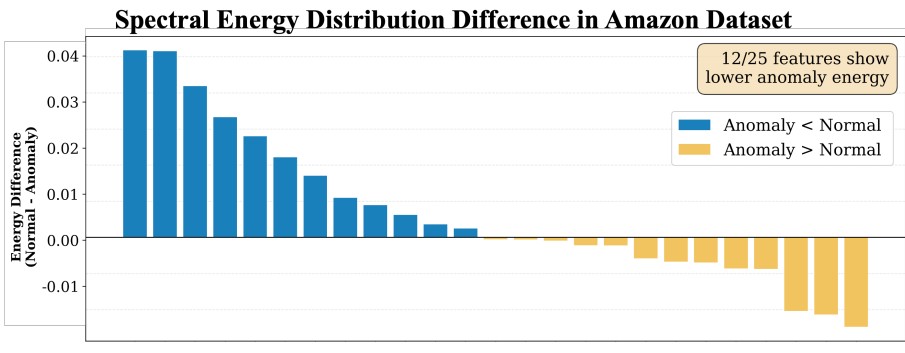

*Figure 6.* Spectral energy distribution in the Amazon dataset. Each bin represents the average spectral energy of all nodes in one feature. Blue indicates normal nodes and yellow indicates anomalies.

## D. Limitations

The proposed method, EGNN, is built upon spectral energy computation. In this work, we focus on node attribute anomalies and treat all edges as unweighted. For the Amazon and YelpChi datasets, we ignore differences in edge connections and assume uniform edge weights. However, in more realistic settings, edges may carry weights, exhibit nonlinear relationships, and be directed. In such cases, the spectral formulation must be reformulated, since the eigenvalues of directed graphs are not guaranteed to be real or positive. EGNN is primarily designed in a semi-supervised setting, as it requires a small amount of anomaly labels to learn the underlying energy shift patterns. In a fully unsupervised scenario, where no anomaly labels are available, the performance of EGNN may degrade, since the model lacks explicit supervision to associate spectral energy variations with anomalous behaviors.

*Table 9.* Performance comparison on static graph datasets (40% training).

| Model | Amazon | | | YelpChi | | | T-Finance | | | T-Social | | |
|---|---|---|---|---|---|---|---|---|---|---|---|---|
| | F1-m | AUC | PRC | F1-m | AUC | PRC | F1-m | AUC | PRC | F1-m | AUC | PRC |
| GCN | 63.22±1.34 | 82.20±1.26 | 34.15±4.28 | 55.70±1.44 | 58.91±2.17 | 22.42±2.69 | 72.63±4.28 | 85.59±2.53 | 43.45±11.75 | 65.34±5.73 | 78.80±13.06 | 23.96±13.61 |
| ChebyNet | 91.03±1.05 | 95.76±1.73 | 86.18±2.32 | 71.59±1.00 | 84.30±0.91 | 54.46±2.54 | 85.70 ±9.88 | 93.29 ±5.74 | 75.37±22.69 | 58.31±0.81 | 73.56±2.21 | 12.04±3.17 |
| GAT | 85.70±4.55 | 93.29±2.10 | 75.37±10.91 | 65.92±0.91 | 78.33±1.05 | 41.34±2.35 | 81.56±5.06 | 91.69±1.71 | 55.90±14.49 | 67.08±0.96 | 85.47±1.81 | 25.65±2.05 |
| GIN | 88.93±1.89 | 92.32±1.85 | 80.29±2.36 | 65.20±3.38 | 76.59±6.42 | 39.51±6.53 | 79.65±1.00 | 84.41±2.84 | 51.83±2.76 | 52.61±2.44 | 66.37±12.05 | 5.92±2.07 |
| GraphSAGE | 75.28±9.26 | 88.90±5.61 | 66.20±16.07 | 68.59±1.80 | 82.17±2.46 | 45.80±4.30 | 57.04±4.68 | 57.98±5.23 | 8.71±4.60 | 58.42±1.02 | 74.04±7.10 | 9.44±0.56 |
| SGC | 63.06±7.38 | 77.10±9.02 | 23.68±6.09 | 51.35±1.35 | 51.91±2.23 | 15.76±1.24 | 57.92±8.53 | 51.91±12.43 | 15.76±13.04 | 42.49±4.78 | 48.93±7.62 | 3.04±0.41 |
| GT | 89.14±3.36 | 90.75±2.21 | 76.20±6.77 | 67.27±1.89 | 79.86±1.84 | 44.89±3.55 | 64.63±6.36 | 78.50±6.97 | 19.87±8.51 | 63.66±4.42 | 83.43±3.66 | 20.82±7.92 |
| BernNet | 91.31±0.60 | 93.83±2.09 | 84.01±2.61 | 69.32±0.61 | 82.14±0.65 | 48.89±1.30 | 81.48±2.85 | 90.66±1.44 | 52.90±9.95 | 51.94±0.96 | 64.68±1.77 | 4.88±1.66 |
| PC-GNN | 67.04±0.85 | 81.80±1.17 | 29.36±4.34 | 56.30±0.25 | 59.55±0.40 | 22.91±1.04 | 85.62±1.19 | 92.17±1.61 | 73.70±3.17 | 51.19±6.20 | 72.88±9.95 | 13.77±11.98 |
| BWGNN | 91.40±0.78 | 96.17±1.09 | 86.42±2.17 | 71.78±1.20 | 84.33±1.15 | 54.67±2.62 | 84.67±9.88 | 93.12±5.74 | 73.84±22.69 | 81.78±2.29 | 94.32±0.81 | 60.81±5.72 |
| UniGAD | 90.46±0.73 | **96.60±0.65** | 86.65±2.21 | 71.23±0.65 | 83.69±0.64 | 53.97±1.63 | 89.34±0.44 | 95.14±0.27 | **84.71±0.48** | 78.67±1.21 | 91.65±0.71 | 58.33± 4.68 |
| **Ours** | **91.52±0.47** | 96.32±0.78 | **89.40±1.10** | **76.89±0.66** | **88.10±0.68** | **66.26±1.52** | **89.60±0.74** | **95.39±0.48** | 84.39±1.51 | **95.40±0.11** | **99.69±0.01** | **95.89±0.13** |

