# OpenReview forum: "Modeling Spectral Energy Shifts in Spatio-Temporal Graph Anomaly Detection"
_ICML.cc/2026/Conference — ICML 2026 regular_

### Official Review · Reviewer_1TUg · 2026-03-07

**Soundness:** 3
**Presentation:** 3
**Significance:** 3
**Originality:** 3
**Overall Recommendation:** 4
**Confidence:** 4

**Summary:**

This paper proposes a node-level anomalies detection method called the Energy Graph Neural Network (EGNN), which uses the gate mechanism to learn left shift and right shift and address the challenge of camouflaged anomalies in both static and time-series graph dynamics, and  achieve SOTA performance compared with other baselines.

**Compliance With Llm Reviewing Policy:**

Affirmed.

**Final Justification:**

After the detailed rebuttal, I think the authors resolve my main concerns, thus I would like to give the final recommendation as "Weak Accept".

**Key Questions For Authors:**

1 The EGNN fused the left-shift and right-shift energy with the gate MLP. The ablation in Table 5 is interesting, which suggests the drop with the left-shift while the datasets only contain right-shift spectrum. Does this imply that the optimal configuration depends on the specific shift pattern in the dataset? If so, how robust is the method when the spectral shift behavior is unknown?

2 Can you perform computational complexity and time complexity analysis for your methods?


Minor:
Table 3 in page 6 occurs before the Table 2 in page 7

**Limitations:**

yes

**Strengths And Weaknesses:**

Soundness:

Good. The paper proposes a well-motivated method with clear objectives. The approach is technically reasonable and the empirical results demonstrate improvements over existing baselines.


Presentation:

Good. The paper clearly motivates the problem of camouflaged anomalies and presents the proposed method in a structured manner.

Significance:

Good. Detecting camouflaged anomalies is an important problem in real-world applications of graph anomaly detection, making this work potentially impactful.

Originality:

Good. The proposed EGNN framework and the spectral-energy modeling perspective appear to be original contributions.

Weakness:

1 Equations (1)–(4) are based on the Rayleigh quotient, but the paper does not sufficiently explain why this formulation is appropriate for modeling anomaly-related spectral shifts here.

2 The paper does not provide a computational complexity analysis. In particular, for time-series graphs, sliding windows along the temporal dimension may significantly increase computational cost. It would be helpful to analyze the time and memory complexity of the proposed approach.

---

> ### Author Rebuttal · Authors · 2026-03-31
>
> We sincerely appreciate the reviewer for the very insightful comments . Below, we address your concerns in detail:
>
> 1. Thank you for this comment. We want to clrify our motivation is that the Rayleigh quotient provides a normalized measure of graph-signal smoothness relative to the topology. For a graph signal $x_f$, $\frac{x_f^\top L x_f}{x_f^\top x_f}$ measures how much energy lies toward higher-frequency components, which directly matches the commonly observed right-shift phenomenon in graph anomaly detection.
>
>     This is suitable because anomaly-related spectral shifts are essentially changes in smoothness. High-variance anomalies make neighboring nodes less consistent, increasing local variation and thus Rayleigh-quotient energy. In contrast, camouflaged anomalies can reduce variation and concentrate energy in lower-frequency components, motivating the complementary left-shift formulation. Our localized version follows the same principle at the neighborhood level, making this spectral measure compatible with message passing while preserving its interpretation as a normalized local energy ratio. Fig. 2 further shows that this local Rayleigh-quotient energy still tracks anomaly-related spectral shifts. We will revise the paper to make this motivation clearer.
>
> | Models       | Amazon | YelpChi | T-Finance | T-Social |
> | ------------ | ------ | ------- | --------- | -------- |
> | GCN          | 1.78   | 2.15    | 5.70      | 69.5     |
> | GraphSage    | 6.78   | 9.77    | 28.27     | 646.0    |
> | GAT          | 67.57  | 61.51   | 298.0     | 1680     |
> | BWGNN        | 4.62   | 6.82    | 19.31     | 497.9    |
> | UniGAD+BWGNN | 5.03   | 8.49    | 20.41     | 659.9    |
> | Ours         | 18.65  | 30.38   | 57.92     | 1990     |
>
> *Table 8: The table of Computation cost measured by GFLOPs.*
>
> 2. The energy computation in Eq. (3) costs $O(EF)$, where $E$ is the number of edges and $F$ is the feature dimension, but it is performed only once as preprocessing. The gating module adds $O(NFG L_g)$, where $N$ is the number of nodes, $G$ is the gate hidden dimension, and $L_g$ is the number of gate layers. Each GraphSAGE layer costs $O(EH + NH^2)$, where $H$ is the hidden dimension. Therefore, the per-epoch training complexity is
> $O(EH + NH^2 + NFG L_g)$. In large graphs typically satisfy $E \gg N$, the message-passing term dominates, so the overall complexity can be approximated as $O(EH)$. We also report the computational cost in Table 8. As graph size increases, our method incurs higher cost, but it consistently achieves the best performance.
>
>
> Q1: We thank the reviewer for this insightful question. Yes, the optimal fusion can depend on the spectral shift pattern of the dataset. If the dataset exhibits mixed or unknown spectral behavior, combining both branches becomes necessary. Our goal is therefore not to assume a fixed shift pattern, but to make the model robust when the underlying spectral behavior is unknown.
>
> This is exactly why we use a learnable gate. The ablation results support this design: using either branch alone is generally less stable, while the gated fusion achieves the best overall performance across datasets. This suggests that the gate can suppress unhelpful components when one shift pattern dominates, while still benefiting from complementary information when both patterns are present.
>
>
> Q2: Please see the response above.
>
> (Thank you for pointing out the figure placement issue. We will correct it in the revised manuscript.)

---

> > ### Author Rebuttal · Reviewer_1TUg · 2026-04-02
> >
> > Thanks for the rebuttal. My concerns are fully solved. I will keep my score since my score has been high already.

---

> > > ### Author Response · Authors · 2026-04-03
> > >
> > > We sincerely thank the reviewer for the feedback and for considering our rebuttal. We also appreciate the reviewer’s support.

---

### Official Review · Reviewer_tMUw · 2026-03-10

**Soundness:** 3
**Presentation:** 3
**Significance:** 2
**Originality:** 2
**Overall Recommendation:** 3
**Confidence:** 4

**Summary:**

The paper argues that current spectral GAD methods overly emphasize right-shift anomalies. In this case, spectral energy moves to high frequencies and overlook camouflaged anomalies that reduce variation and induce “left-shift” toward low frequencies. This work proposes EGNN, which can compute node-level local spectral energy. This GNN introduces a left-shift energy concept via a flipped Laplacian, and uses a learnable gate to combine left/right energy before message passing. The temporal extension applies the same energy pipeline in sliding windows without explicit sequence modules. Experiments on 4 static and 3 time-series datasets show strong performance.

**Compliance With Llm Reviewing Policy:**

Affirmed.

**Final Justification:**

the rebuttal addressed my major concern. According to the overall quality of this paper. I maintain my original score.

**Key Questions For Authors:**

Q1: Is G actually binary during training or continuous in [0,1]? The problem is that, if it is binary, what threshold is used and how do you backpropagate? If it is adaptable, why not add an ablation comparing binary against continuous gates.

Q2:How are edges constructed for SWaT, WADI, and MSL? Are they static or encoded per window? Do these edges change over time?

Q4:The authors claim robustness to longer windows. Robustness is related to the performace under abnormal environemnts. Can you include a sensitivity study for three temporal datasets.

**Limitations:**

Yes

**Strengths And Weaknesses:**

S1: The paper points out an under-discussed problem about camouflaged anomalies and gives a spectral interpretation for left-shift phenomenon due to distributional collapse. The qualitative evidence on YelpChi and Amazon supports that mixed left and right patterns exist.

S2: The local Rayleigh-quotient formulation Eq. 2 and its efficient per-node surrogate are compatible with message passing and practical at scale. The energy-only preprocessing is conceptually simple and broadly applicable to very large graph datasets.

S3: This is an unified model for both static and temporal features. The proposed time-series variant avoids RNN, such as Transformer encoders. This design can improve efficiency and stability for long windows.

Weakness:

W1: The paper does not present new theorems or formal guarantees. The formulations mostly restate Rayleigh quotient energies locally and define a flipped Laplacian energy. These are mathematically correct and clearly presented. There are no generalization bounds or approximation guarantees for the local surrogate beyond experiments.

W2: Only one ablation study is shown. There is no study of window length robustness, gate architecture, neighborhood size, or energy normalization variants. Altgough, Table 5 also shows that adding left-shift can slightly hurt when right-shift dominates. Why? Is this results showing the gate fails to fully adapt?

W3: Section 3.6 says GraphSAGE performs spatial aggregation per time step and the gate captures temporal dependencies implicitly. The paper should specify how information across time within the window is aggregated to make a prediction. In addition, how labels are aligned with the windows?

---

> ### Author Rebuttal · Authors · 2026-03-31
>
> We sincerely appreciate the reviewer for the very insightful comments . Below, we address your concerns in detail:
>
> 1. We have addressed this concern in our response to Reviewer 7zMX (3).
>
> 2. To address this concern, we conducted several ablation studies to validate our design choices. Specifically, Table 3 presents the gate design and component ablations, Table 6 studies the hop size, and Table 7 evaluates the window length. As the hop size increases, local spectral energy becomes less discriminative. In the extreme case, when the hop size covers the graph diameter, every node recovers the same global spectral energy, losing node-level discrimination. Although 3-hop achieves the best overall score in Table 6, the improvement over 1-hop is marginal while incurring substantially higher computational cost. Considering the trade-off between effectiveness and efficiency, we choose 1-hop as the more practical design. Across all these ablations, the results consistently show that our selected hyperparameters and components provide the best balance between effectiveness and efficiency.
>
>     Regarding the slight performance drop observed when incorporating left-shift spectral energy in the previous ablation, we believe this is because the anomalies in the two large datasets, T-Finance and T-Social, predominantly exhibit right-shift behavior. In other words, these anomalies are less likely to be camouflaged and instead mainly correspond to high-variance patterns. Under this setting, the left-shift component provides limited additional benefit. Since the gate is learned rather than fixed, it may not converge to a perfect assignment of 0 for left-shift and 1 for right-shift in every case, which can lead to a slight performance decrease.
>
> | Setting    | Amazon    |           |           | YelpChi   |           |           |
> | ---------- | --------- | --------- | --------- | --------- | --------- | --------- |
> |            | F1-m      | AUC       | PRC       | F1-m      | AUC       | PRC       |
> | 1-hop      | 91.80     | 96.88     | 90.33     | 76.91     | 88.12     | 66.10     |
> | 3-hop only | **92.18** | 96.17     | **89.18** | **78.21** | **88.74** | **67.41** |
> | 5-hop only | 92.02     | **96.33** | 89.16     | 77.35     | 88.11     | 66.82     |
> | 7-hop only | 89.97     | 96.31     | 84.73     | 74.36     | 87.84     | 63.07     |
>
> *Table 6: Ablation for differnt hop size.*
>
> | Window Size | WADI  |       |       | MSL   |       |       |
> | ----------- | ----- | ----- | ----- | ----- | ----- | ----- |
> |             | F1-m  | AUC   | PRC   | F1-m  | AUC   | PRC   |
> | 5           | 67.01 | 78.74 | 36.91 | 71.41 | 83.61 | 95.63 |
> | 10          | 65.02 | 75.72 | 29.39 | 70.96 | 83.86 | 95.81 |
> | 15          | 68.37 | 75.87 | 35.87 | 72.89 | 86.07 | 96.51 |
> | 30          | 66.77 | 77.97 | 32.62 | 70.54 | 86.66 | 96.91 |
>
> *Table 7: Ablation study for different window sizes.*
>
> 3. Thank you for pointing out this ambiguity. We want to clarify that, given a sliding window of length $w$ each node is represented by a w-dimensional feature vector containing its sensor readings over consecutive time steps. The temporal dimension is treated as the feature dimension, so the energy computation, gating, and normalization operate directly along this axis to capture each node’s spectral energy profile over the full window. GraphSAGE then performs spatial aggregation across nodes, and a global mean pooling over all nodes yields the final window-level representation for classification. In this way, temporal information is handled implicitly through windowed node features, while spatial information is modeled explicitly through message passing. Each window is assigned a binary label: it is labeled anomalous if any time step within the window contains an anomaly, and normal otherwise. We will include these implementation details explicitly in the revised manuscript.
>
>
> Q1: The gate output $G$ is continuous in $[0,1]$. We also provide an ablation study of this component; please refer to our response to Reviewer VF5s.
>
> Q2: For SWaT, WADI, and MSL, we use a fully connected static graph, where each sensor is a node and edges connect all sensor pairs. The topology is constructed once and shared across all sliding windows. Temporal information is encoded in the node features and the spectral energy computation, which yields window-specific energy ratios over time.
>
> Q3: The corresponding results are reported in Table 7.

---

> > ### Author Rebuttal · Reviewer_tMUw · 2026-04-03
> >
> > please refer to my comments. And I currently prefer to maintain my original score.

---

### Official Review · Reviewer_VF5s · 2026-03-12

**Soundness:** 2
**Presentation:** 2
**Significance:** 2
**Originality:** 3
**Overall Recommendation:** 4
**Confidence:** 3

**Summary:**

EGNN proposes a spectral energy framework for graph anomaly detection that extends prior right-shift formulations to also capture left-shift anomalies, whose energy concentrates in low frequencies due to distributional collapse. It computes localized Rayleigh quotient energy at the 1-hop level, derives a left-shift counterpart via a flipped Laplacian, and fuses both via an adaptive gating MLP before classification with GraphSAGE. The framework extends to time-series graphs via sliding-window energy computation without explicit temporal modules.

**Compliance With Llm Reviewing Policy:**

Affirmed.

**Final Justification:**

Thanks to the authors' rebuttal, my concerns have been fully addressed. I'm very willing to raise my rate.

**Key Questions For Authors:**

Please refer to Weaknesses.

**Limitations:**

yes.

**Strengths And Weaknesses:**

**Strengths:**

1)	The paper is easy to follow.
2)	The paper is well written.
3)	The paper seems novel to me.

**Weaknesses:**

1) The gating mechanism uses a simple feature-wise linear interpolation between left- and right-shift energies. No justification is provided for this design choice over alternatives such as concatenation, attention, or bilinear fusion, and no comparison against these strategies is included.
2) Comparisons against label-efficient baselines or unsupervised methods across varying label rates (e.g., 1%, 5%, 10%, 20%) should be supplemented.
3) The left-shift energy formulation $E^{(L)} = 2 − E^{(R)}$ follows directly from substituting the flipped Laplacian $2I – L$ into the Rayleigh quotient. This is a standard spectral identity and does not constitute an independent technical contribution.
4) More recent methods specifically designed for time-series graph anomaly detection are absent, which undermines the validity of the state-of-the-art comparison.
5) The gating MLP takes raw node features $X$ as input to decide how to combine left- and right-shift energies. If $X$ alone is sufficient to determine the anomaly type, the energy transformation step is redundant. This circular dependency undermines the motivation for the proposed architecture.
6) The ablation study in Table 5 only evaluates the presence or absence of the left-shift component. Lack fixed equal-weight mixing versus adaptive gating, raw features fed directly into GraphSAGE without energy transformation; and alternative backbones such as GCN or GAT in place of GraphSAGE.
7) The paper fixes the neighborhood at 1-hop without evaluating the effect of varying the aggregation radius. A comparison between 1-hop and 2-hop local energy would clarify whether the localization choice is principled or arbitrary.
8) No runtime or computational comparison.

---

> ### Author Rebuttal · Authors · 2026-03-31
>
> We sincerely thank the reviewer for the insightful feedback, which helped improve our paper. We address each concern below.
>
>
>
> 1. We added a component ablation in Table 3 comparing the MLP gate with cross-attention and bilinear fusion. The MLP gate performs best on both datasets. We believe this is because it only needs to balance left- and right-shift spectral energies feature-wise, so more complex designs bring little benefit. Cross-attention is likely redundant since local spectral energy already captures neighborhood information, while bilinear fusion adds parameters without improving performance. These results support our simple MLP gate.
>
> | Setting                    | Amazon    |           |           | YelpChi   |           |           |
> | -------------------------- | --------- | --------- | --------- | --------- | --------- | --------- |
> |                            | F1-m      | AUC       | PRC       | F1-m      | AUC       | PRC       |
> | MLP Gate + GCN             | 75.40     | 87.84     | 49.68     | 58.13     | 63.68     | 27.44     |
> | MLP Gate + GAT             | 60.03     | 62.76     | 19.88     | 62.55     | 73.33     | 35.25     |
> | Cross-Attention+ GraphSAGE | 89.35     | 95.16     | 84.55     | 75.19     | 86.49     | 62.34     |
> | Bilinear  + GraphSAGE      | 88.78     | 93.62     | 82.24     | 72.80     | 84.74     | 57.84     |
> | Original                   | **91.80** | **96.88** | **90.33** | **76.91** | **88.12** | **66.10** |
>
> *Table 3: Ablation study of model components.*
>
>
>
> 3. We also evaluated different label ratios. Due to space limits, we report only the 10% label setting in Table 4. Under limited labels, BWGNN performs better on Amazon, but our method achieves the best overall results.
>
> | Model     | Amazon(10%) |           | YelpChi(10%) |           | T-Social(10%) |           |               |
> | --------- | ----------- | --------- | :----------- | --------- | ------------- | --------- | ------------- |
> |           | F1-m        | PRC       | F1-m         | PRC       | F1-m          | PRC       | Average(F1-m) |
> | GCN       | 63.22       | 34.15     | 57.09        | 24.94     | 62.32         | 21.20     | 60.87         |
> | GraphSage | 72.87       | 65.18     | 61.12        | 42.65     | 61.18         | 13.62     | 65.05         |
> | BernNet   | 89.98       | 87.76     | 69.37        | 48.92     | 54.65         | 6.26      | 71.33         |
> | BWGNN     | **90.75**   | **88.88** | 71.66        | 54.42     | 72.09         | 42.25     | 78.17         |
> | Ours      | 87.18       | 82.92     | **71.77**    | **55.46** | **96.15**     | **96.64** | **85.03**     |
>
> *Table 4: Benchmark for the 10% label setting.*
>
> 3. We want to clarify that Eq. (4) can be directly derived from the flipped Laplacian and is not, by itself, an independent technical contribution. Our contribution lies in the overall framework: we identify a left-shift phenomenon in real benchmarks, show that prior right-shift spectral methods miss this behavior, and develop a node-level spectral formulation with adaptive fusion to make this signal usable in message-passing GNNs. The ablations further show that explicitly modeling left-shift energy improves detection when mixed left-/right-shift behavior exists.
>
> 4. We have added the recent method [1] in Table 5 and evaluated it on MSL and WADI. The results show that our method remains the strongest overall.
>
> | Models     | MSL(F1-m)   | WADI(F1-m)  |
> | ---------- | ----- | ----- |
> | GeneralDyG | 67.78 | 59.73 |
> | Ours       | **70.47** | **61.57** |
>
> *Table 5: Additional evaluation on temporal datasets.*
>
> 5. The gating MLP does not perform anomaly detection; it only predicts a mixing coefficient in $[0,1]$ to interpolate between left- and right-shift energies, which is much simpler than classification. The raw features $X$ provide node-specific routing context, while the anomaly-relevant signal is captured by the spectral energy representations, which $X$ alone cannot encode. As shown in Table 1, GraphSAGE with raw features suffers a big performance drop, confirming that the spectral energy transformation is necessary rather than redundant.
>
> 6. We added a backbone ablation in Table 3 with GCN and GAT. Replacing GraphSAGE with either one consistently hurts performance. We use GraphSAGE because it preserves self-information during neighbor aggregation. By contrast, GCN adds another spectral-style propagation that may be redundant and distort spectral semantics, while GAT introduces extra attention-based reweighting that appears unnecessary after the MLP gate selects important energy features.
>
> 7. We have addressed a similar question in our response to Reviewer tMUw (2).
>
>
>
> 8. We thank the reviewer for raising the runtime concern. We have addressed this in our response to Reviewer 1TUg (2).
>
> [1] Yang, Xiao, Xuejiao Zhao, and Zhiqi Shen. "A generalizable anomaly detection method in dynamic graphs." Proceedings of the AAAI Conference on Artificial Intelligence, 39(20), 2025.

---

> > ### Author Rebuttal · Reviewer_VF5s · 2026-04-05
> >
> > Thanks to the authors' rebuttal, my concerns have been fully addressed. I'm very willing to raise my rate.

---

> > > ### Author Response · Authors · 2026-04-05
> > >
> > > We sincerely thank the reviewer for the feedback and for considering our rebuttal. We also appreciate the reviewer’s support.

---

### Official Review · Reviewer_7zMX · 2026-03-13

**Soundness:** 2
**Presentation:** 2
**Significance:** 2
**Originality:** 2
**Overall Recommendation:** 4
**Confidence:** 2

**Summary:**

The paper shows that this type of anomaly persists across multiple datasets and remains undetectable by existing spectral approaches. To address this limitation, we propose a node-level spectral energy formulation that is fully compatible with message passing and enables the detection of camouflaged anomalies. Building on this formulation, we introduce an energy-aware graph learning framework that models spectral shifts through energy-driven message passing in both static and time-series graphs. Besides, this paper unified architecture extends to temporal settings without introducing specialized sequence modules, enabling efficient learning under long sliding windows. Extensive experiments on large-scale benchmarks demonstrate the effectiveness and scalability of this approach.

**Compliance With Llm Reviewing Policy:**

Affirmed.

**Final Justification:**

Based on the scores from other reviewers and the authors' responses, I increased my score but decreased my confidence level.

**Key Questions For Authors:**

1. Figures 1 and 4 only show group-level mean differences. Since normal nodes already concentrate ~80% of their energy in the low-frequency region, the paper does not explain how the model distinguishes naturally smooth normal nodes from camouflaged anomalies within the same low-frequency range. Node-level distribution visualization and separability analysis are needed.

2. The synthetic experiments in Figure 2 only inject high-variance anomalies (σ > 1), which validates the right-shift phenomenon — a result already established by prior work such as BWGNN. The paper's core contribution is the left-shift phenomenon, yet no symmetric experiment injecting low-variance camouflage anomalies (σ < 1) is provided. Figure 2 does not support the paper's own key claim.

3.  The paper approximates global spectral energy using 1-hop local subgraphs but provides no theoretical error bound. By contrast, the Chebyshev approximation cited in the same section comes with well-established approximation guarantees. The proposed local approximation resembles a heuristic, and its reliability on sparse or heterophilic graphs remains unknown.

4. The strongest baseline, BWGNN, dates from ICML 2022, and most other baselines are from 2020–2021. The paper cites UniGAD (NeurIPS 2024) in the related work but does not include it as a baseline, with no explanation given. This omission needs to be justified.

5. The ablation in Table 5 cannot rule out that the performance gain from E^((L)) comes from increased feature dimensionality rather than spectral semantics. A controlled experiment replacing E^((L)) with dimension-matched but semantically irrelevant features (e.g., random walk statistics) on YelpChi and Amazon is necessary to validate the claimed contribution of left-shift energy.

**Limitations:**

No,see Questions

**Strengths And Weaknesses:**

Advantages:

Well-written and easy to follow

Clearly motivated

Disadvantages:

Outdated baselines

Lack of ablation studies

---

> ### Author Rebuttal · Authors · 2026-03-31
>
> We sincerely thank the reviewer for the insightful comments and address them below.
>
> 1. EGNN does not rely on absolute low-frequency energy. Instead, it distinguishes camouflaged anomalies by their stronger low-frequency concentration captured by $E^{(L)}$. As shown in Fig. 1, although both normal and camouflaged anomalous nodes are low-frequency dominated, camouflaged anomalies have lower spectral energy in 26 of 32 features on YelpChi. Distribution collapse further suppresses variance, pushing $E^{(R)}$ toward zero, while $E^{(L)} = 2 - E^{(R)}$ amplifies this subtle gap.
>
> 2. We would like to clarify that, although BWGNN uses a visually similar plot, the purpose is different. BWGNN uses it to show that global spectral energy captures the right-shift phenomenon. Our Fig. 2 instead shows that local spectral energy can also capture the right-shift phenomenon, demonstrating the effectiveness of the local formulation.
>
>
>
> 3. Our method is evaluated on relatively dense and homophilic graphs, where local spectral modeling is more reliable. In particular, by the eigenvalue interlacing theorem [1], if $H$ is an induced subgraph of $G$ with $m$ vertices, then its eigenvalues are bounded by those of $G$:
> $$
> \lambda_k(G)\le \lambda_k(H)\le \lambda_{k+n-m}(G).
> $$
> This shows that the local subgraph spectrum remains constrained by the global spectrum, rather than being arbitrary. We further verify this gap empirically on synthetic homophilic Barabási–Albert graphs (n=1000) with varying density. As shown in Table 0, we compute the average gap of this error bound. The gap is much larger on sparse graphs, but becomes very small on dense graphs. Since our benchmarks are substantially denser, the proposed local $1$-hop approximation is applied in a favorable regime.
>
> | Avg Degree | Error Bound Gap (%) |
> | ---------- | ------- |
> | 2          | 82.24  |
> | 10         | 55.77  |
> | 20         | 39.87   |
> | 48         | 23.38    |
> | 90         | 14.79    |
> | 160        | 8.16    |
>
> *Table 0: Local vs. Global spectral energy on BA Graphs.*
>
>
>
> 3. We thank the reviewer for highlighting UniGAD [2]. Although we were aware of it, we did not include it initially because it is designed for node-, edge-, and graph-level anomaly detection, while our benchmark focuses on node anomaly detection. We have added UniGAD+BWGNN in Table 1. The results show that UniGAD is a strong baseline, achieving the best AUROC on Amazon, while our method remains highly competitive overall.
>
> | Model        | Amazon    |           | YelpChi   |           | T-Social  |           |          |
> | ------------ | --------- | --------- | --------- | --------- | --------- | --------- | -------- |
> |              | F1-m      | AUC       | F1-m      | AUC       | F1-m      | AUC       | Time (s) |
> | GCN          | 62.16     | 81.20     | 56.96     | 60.71     | 63.67     | 88.89     | 2041     |
> | GraphSage    | 74.66     | 88.19     | 71.64     | 84.99     | 59.15     | 75.23     | 3063     |
> | GAT          | 86.93     | 94.81     | 67.87     | 84.43     | 71.76     | 87.18     | 2945     |
> | BWGNN        | 91.73     | 96.24     | 71.57     | 84.26     | 82.47     | 95.31     | 3045     |
> | UniGAD+BWGNN | 89.94     | **96.93** | 72.47     | 86.78     | 78.67     | 91.65     | 3981     |
> | Ours         | **91.80** | 96.88     | **76.91** | **88.12** | **95.27** | **99.14** | 3646     |
>
> *Table 1: Additional evaluation including running time.*
>
>
> 4. We report additional ablations in Table 2, comparing the full model with variants using only one type of energy, fixed equal-weight fusion, and a semantically irrelevant random-walk feature (SRW). The results show that: (1) either spectral branch alone is insufficient because Amazon and Yelp exhibit mixed spectral patterns; (2) learnable fusion is necessary, as fixed equal-weight fusion degrades performance on both datasets; and (3) the benefit of left-shift energy is not due to increased dimensionality, since replacing it with SRW lowers performance.
>
> | Setting         | Amazon    |           | YelpChi   |           |
> | --------------- | --------- | --------- | --------- | --------- |
> |                 | F1-m      | AUC       | F1-m      | AUC       |
> | $E^{(R)}$  only | 87.20     | 94.97     | 72.21     | 83.61     |
> | $E^{(L)}$  only | 87.73     | 95.08     | 71.77     | 83.47     |
> | Fixed           | 72.86     | 86.27     | 49.44     | 50.22     |
> | SRW             | 74.23     | 88.04     | 50.31     | 50.67     |
> | Original        | **91.80** | **96.88** | **76.91** | **88.12** |
>
> *Table 2: Ablation study on the effectiveness of left-shift spectral energy.*
>
>
> [1]Haemers, W. H. (1995). Interlacing eigenvalues and graphs. Linear Algebra and its applications, 226, 593-616.
>
> [2]Lin, Yiqing, et al. "Unigad: Unifying multi-level graph anomaly detection." Advances in neural information processing systems 37 (2024): 136120-136148.

---

> > ### Author Rebuttal · Reviewer_7zMX · 2026-04-02
> >
> > While I appreciate the authors' efforts to provide additional experiments during the rebuttal phase, I must maintain a cautious stance. The added visualizations (t-SNE) and a single point synthetic test ($\sigma=0.5$) are somewhat anecdotal and do not fully prove the robustness of the 'left-shift' claim across continuous distributions. Furthermore, redefining the spectral approximation as an 'exact local spectrum' sidesteps my concern regarding the expressiveness of 1-hop subgraphs on highly heterophilic graphs. Overall, the method has potential, but the empirical rigor requires substantial fortification.

---

> > > ### Author Response · Authors · 2026-04-06
> > >
> > > Table 9: Homophily Analysis for Graph Anomaly Detection Datasets
> > >
> > > Adjusted Homophily: $h_{adj} = \frac{h_{edge} - h_{rand}}{1 - h_{rand}} $
> > >
> > > where $h_{rand} = \sum_k p_k^2 $ is the expected same-label edge ratio under random connections.
> > >
> > > | Dataset   | $h_{edge} $ | $h_{node} $ | $h_{rand} $ | $h_{adj} $ |    Regime    |
> > > | --------- | :---------: | :---------: | :---------: | :--------: | :----------: |
> > > | Amazon    |   0.9543    |   0.9098    |   0.8720    |   0.6432   |  Homophilic  |
> > > | T-Finance |   0.9708    |   0.9566    |   0.9125    |   0.6660   |  Homophilic  |
> > > | Yelp      |   0.7731    |   0.7699    |   0.7516    |   0.0865   | Near-neutral |
> > > | T-Social  |   0.6239    |   0.8778    |   0.9415    |  −5.4316   | Heterophilic |
> > >
> > >
> > >
> > > We agree that a formal guarantee for the local spectral energy approximation is currently unavailable and remains an important open problem. Our goal in this work is therefore not to claim theoretical optimality, but to demonstrate that this approximation is practically useful for graph anomaly detection. The design is attractive because it is computationally efficient, captures anomaly patterns in the spectral energy domain, and can be seamlessly incorporated into standard MPNN architectures. Empirically, it delivers **strong and consistent gains across benchmarks**. Notably, on T-Social, our method **clearly surpasses all competing baselines**, and on the public GAD benchmark [1], its performance is already **close to the upper-bound XGBoost** baseline in both AUC and PRC.
> > >
> > > For the specific concern about 1-hop approximation on heterophilic graphs, we further analyze dataset homophily statistics following [2]. Besides edge and node homophily, we also report adjusted homophily to account for severe class imbalance. Table 9 shows that T-Social is strongly heterophilic. Despite this challenging setting, our method still achieves the best quantitative performance. **This suggests that, while the approximation is heuristic at present, it is robust in practice across substantially different homophily regimes.**
> > >
> > > [1] Lim, D., Hohne, F., Li, X., Huang, S. L., Gupta, V., Bhalerao, O., & Lim, S. N. (2021). Large scale learning on non-homophilous graphs: New benchmarks and strong simple methods. *Advances in neural information processing systems*, *34*, 20887-20902.
> > >
> > > [2] Tang, J., Hua, F., Gao, Z., Zhao, P., & Li, J. (2023). Gadbench: Revisiting and benchmarking supervised graph anomaly detection. *Advances in Neural Information Processing Systems*, *36*, 29628-29653.

---

### Decision · Program_Chairs · 2026-04-30

**Decision:**

Accept (regular)

**Comment:**

This paper investigates camouflaged anomalies via left-shift spectral energy and proposes an energy-aware GNN (EGNN) that jointly models left- and right-shift behaviors. The problem is well-motivated, and the proposed framework is simple, scalable, and compatible with message passing. Empirical results on static and temporal benchmarks show consistent improvements, and the rebuttal strengthens the paper with additional ablations, baselines, and analysis.

The reviewers generally agree on the clarity, relevance, and practical potential of the work, and several concerns were addressed during rebuttal (e.g., additional ablations and comparisons).

That said, several remaining concerns raised by reviewers should be acknowledged:

1. R7zMX questions the empirical support for the left-shift claim, noting limited validation (e.g., lack of systematic low-variance synthetic experiments and insufficient node-level separability analysis), and raises concerns about missing strong baselines and limited ablations.
2. RVF5s points out that parts of the formulation rely on standard spectral identities, and highlights insufficient comparisons (e.g., recent methods, label-efficient settings) and limited justification of design choices such as the gating mechanism. Also, some following questions were not answered by the authors.
3. RtMUw emphasizes the lack of theoretical guarantees for the local spectral approximation and calls for more comprehensive ablations (e.g., window size, hop size, robustness analysis), some of which remain only partially resolved.
4. R1TUg notes missing complexity analysis and clearer justification of the spectral formulation, though these are largely addressed in the rebuttal.

Overall, while the paper has some limitations in theoretical grounding and empirical completeness, it introduces a useful perspective on spectral anomaly modeling and demonstrates promising performance. Considering the novelty of modeling left-shift anomalies and the strengthened experimental evidence after rebuttal, I recommend weak accept.